# Analysis of Predisposing Factors for Rapid Dental Calculus Formation

**DOI:** 10.3390/jcm9030858

**Published:** 2020-03-20

**Authors:** Carla Fons-Badal, Antonio Fons-Font, Carlos Labaig-Rueda, M. Fernanda Solá-Ruiz, Eduardo Selva-Otaolaurruchi, Rubén Agustín-Panadero

**Affiliations:** Department of Oral Medicine, Faculty of Medicine and Dentistry, University of Valencia, 46010 Valencia, Spain; carlafonsbadal@gmail.com (C.F.-B.); labaig@uv.es (C.L.-R.); m.fernanda.sola@uv.es (M.F.S.-R.); Eduardo.j.selva@uv.es (E.S.-O.); rubenagustinpanadero@gmail.com (R.A.-P.)

**Keywords:** dental calculus, heavy calculus formers, oral bacteria

## Abstract

Background: Calculus accumulation varies widely between individuals. Dental calculus has been associated with the principal periodontal diseases. The aim of this study was to analyze individual characteristics, and salivary and microbiological parameters among patients considered to be rapid calculus formers and patients who form calculus slowly. Methods: Individual characteristics were recorded in a sample of 74 patients (age, sex, smoking, periodontal diagnosis, and dental crowding), as well as salivary parameters (unstimulated saliva flow, pH, and biochemical analysis of saliva) and microbiological parameters (by means of semi-quantitative polymerase chain reaction (PCR) analysis). Results: A statistically significant association (*p* = 0.002) was found between the rate of calculus formation and the diagnosis of periodontal disease. A greater presence of dental crowding was observed among the group of rapid calculus formers. Urea and phosphorus levels were higher among rapid calculus formers. Regarding microbiological parameters, differences were found in *Streptococcus mutans*, this being higher in the group of slow formers. Conclusions: Rapid calculus formation appears to be linked to patients diagnosed with more severe periodontal diseases. Rapid calculus-forming patients present more dental crowding and a lower proportion of *S. mutans*.

## 1. Introduction

The main periodontal diseases, gingivitis and periodontitis, have been associated with the accumulation of dental calculus. While it was believed that the accumulation of dental calculus was the main cause of periodontal disease, numerous in vivo studies have shown that biofilm is the determining factor in these diseases [1]. 

Dental calculus is the consequence of the mineralization of bacterial plaque. It constitutes a major oral health problem as it increases the accumulation of plaque and bacterial toxins and impedes their adequate elimination due to surface roughness. This also presents an obstacle to effective hygiene maintenance, which in turn facilitates further plaque formation [2].

A series of factors are related to calculus formation. Those of note include factors related to increases in bacterial plaque retention [3], biochemical factors such as the composition of saliva or crevicular fluid [4,5], factors associated with the microorganisms that compose bacterial plaque [6,7,8], and factors related to diet [9].

It is known that dental calculus formation is not uniform across the whole population; accumulation varies widely from individual to individual. Some subjects, despite maintaining good plaque control, accumulate calculus very quickly. This means that they require more frequent visits to the dentist and dealing with periodontal pathology becomes more difficult. 

The aim of this study is to analyze differences between patients considered rapid calculus formers compared with patients who form calculus slowly. The specific aims are to analyze individual characteristics among patients (age, sex, smoking, periodontal diagnosis, and dental crowding) and to analyze salivary and microbiological parameters in order to determine whether there are significant relations between these factors and calculus formation. 

## 2. Materials and Methods

This investigation was approved by the University of Valencia Ethics Committee and met the Declaration of Helsinki and European Council guidelines for research involving human subjects, as well as Spanish legislation applying to biomedical research, data protection, and bioethics. The sample was recruited from patients attending the University of Valencia dental clinic. All had antecedents of periodontitis, which had been treated, so that patients were now in the maintenance phase, with periodontal pockets ≤ 3 mm, a Silness and Löe plaque index ≤ 1 (plaque is not visible but can be wiped off with an explorer [10]), and a Löe and Silness gingival index ≤ 1 (mild inflammation, slight color change, mild alteration of the gingival surface, no bleeding [11]). The sample was divided into two groups, rapid calculus formers and slow calculus formers. To evaluate whether an individual patient was a rapid calculus former or not, it was necessary to observe her/his evolution. An individual could not be classified as a rapid former simply because she/he presented a large amount of calculus at a single moment, as this could be the outcome of accumulation over many years with no prophylaxis, rather than constituting a case of rapid formation. However, over 8 weeks, a rapid-forming subject may accumulate a Volpe–Manhold index score of over 7, as there are wide variations between individuals [12]. The Volpe–Manhold index is a method to quantify calculus formation measurements on lingual surfaces of anterior lower teeth using a periodontal probe [13]. Rapid calculus formers, despite maintaining good plaque control (Silness and Löe plaque index ≤ 1), need frequent periodontal maintenance. The control group was made up of patients who maintained good plaque control and formed little calculus.

The study sample included 74 patients, of whom 40 (54.1%) were rapid formers and 34 (45.9%) slow formers, with a mean age of 51.3 ± 9.2 years; 53 (71.6%) were women and 21 (28.4%) were men. Most of the sample were not smokers (63.5%, *n* = 47).

## 3. Data Collection

Firstly, patients’ individual characteristics were recorded: age, sex, and smoking habits (number of cigarettes per day). A periodontal examination was performed to assess periodontitis stability and make an initial diagnosis of periodontal disease according to the classification of Caton et al. 2018 [14]. The presence of factors affecting bacterial plaque retention such as dental crowding was also recorded. Dental crowding can be defined as a discrepancy between tooth and jaw sizes that results in malposition and/or rotation of teeth [15]. Its presence or absence was measured only at the fifth sextant because this was where we were going to measure the index. 

To record salivary parameters, an unstimulated saliva flow test was conducted as the flow rate in repose is more stable than under stimulation. To do this, saliva samples were collected with the drainage technique, which consists of letting saliva drop spontaneously from the patient’s mouth into a graduated test tube via a sterile plastic funnel for a fixed time period (10 min). The patients were given a set of instructions before saliva collection: to not eat or chew gum, brush their teeth, or smoke for two hours before collection. Saliva collection was performed in a quiet setting early in the morning to avoid external stimuli and to minimize variations. Each patient was seated, eyes open, with the head slightly inclined forwards to avoid orofacial movement [16].

When 10 min had passed, the volume collected was recorded and the quantity of saliva in milliliters per minute calculated. Then, Eppendorf tubes were filled with saliva with a pipette for chemical analysis, which was performed at La Fe Hospital (Valencia, Spain).

A digital pH meter (PCE, Albacete, Spain), which provides a reading of within 0.01 pH and a precision of ±0.02 pH, was used to determine pH. The range for pH measurement is 0 to 14; pH was measured immediately after saliva collection so that carbon dioxide loss would not modify the reading.

To analyze microbiological parameters, a plaque sample was collected from the fifth sextant of each patient. The plaque was collected from the lingual and vestibular faces of the teeth with a sterile curette and placed in an Eppendorf tube with a conservation solution (reduced transport fluid (RTF)) for later analysis.

All samples were sent to a laboratory (Laboral, Barcelona, Spain) for semi-quantitative polymerase chain reaction (PCR) analysis to determine the presence and proportion of: *Eubacterium saburreum, Corynebacterium matruchotii, Veillonella parvula, Streptocoucus salivarius, Streptocoucus sanguis* and *Streptococus mutans.* According to various studies, these microorganisms are the calcifying species that predominate in supragingival calculus [6,7,17].

For statistical analysis of the data obtained, a logistic regression model was estimated for the “speed of calculus formation” variable. Independent variables were those that were statistically significant in bivariate tests (*p* < 0.05) or that showed a strong statistical tendency (*p* < 0.1). The significance level employed was 5% (α = 0.05). For tests such as the *t*-test for independent samples, with a significance level of 5% and considering an effect size of 0.65, the power reached in the test was 0.78 for the sample size *n* = 74.

## 4. Results

The complete sample was divided into two groups according to the speed of calculus formation (Figure 1).

Analysis of individual characteristics (sex, age, smoking) did not reveal any significant differences between the two groups. A statistically significant association was found (*p* = 0.002) between the rate of calculus formation and periodontal diagnosis. This relation between rapid formers and the type of periodontitis they presented is shown in Figure 2, whereby Stages III and IV and Grade C were associated with rapid calculus formation (advanced and aggressive periodontitis according to the classification of Armitage 1999 [18]).

The frequency of dental crowding (presence or absence) also exhibited a significant difference between the groups (*p* = 0.011). A greater presence of overcrowding was recorded in the rapid formation group (64.7%) in comparison with the group of slow calculus formers (35.3%) (Figure 3).

Table 1 shows the results for salivary parameters, with statistically significant differences between groups for urea (*p* < 0.001) and phosphorus (*p* < 0.001) levels, whereby rapid calculus formers presented higher levels. Uric acid levels were also different between groups, being higher in the rapid formation group, although this difference did not reach statistical significance (*p* = 0.083) (Figure 4).

Data obtained for pH and saliva flow did not reveal any significant differences between groups.

PCR analysis of microbiological parameters found a positive percentage of different bacteria in both groups. In other words, all the bacteria analyzed were present in both groups, although in different proportions (Figure 5).

A notable difference between the groups was only found for *S. mutans*, which was higher in the group of slow calculus formers (slow formers: 58282.6 colony-forming units (CFU); fast formers: 4801.8 CFU), with a statistically significant difference (*p* < 0.001).

There was no overall difference in the level of bacterial load between the groups, as shown in the following graph (Figure 6).

Using the data obtained in the various analyses, a logistic regression model was created to estimate the likelihood that an individual subject would belong to one group or the other according to the factors and parameters taken together. In other words, the model aimed to identify prognostic factors that alone would assign subjects to the group of fast or slow calculus formers:The presence of *S. mutans* was significantly associated (*p* = 0.046) with the probability of accumulating less calculus.Phosphorus was directly associated with the speed of calculus formation (more phosphorus was found in rapid formers). However, the relation between the level of phosphorus and the probability of accumulating calculus differed depending on whether or not the subject presented dental crowding (significant interaction: *p* = 0.027) (Figure 7). For example, if crowding was present, a small increase in phosphorus resulted in a sharp increase in the probability of calculus accumulation. When dental crowding was not present, the slope of the curve in the 10–30 range was less steep and showed less fit.Levels of urea also showed a significant interaction (*p* = 0.015), although the pattern was less clear (Figure 8).

If the individual did not present dental crowding, the probability of belonging to the group of rapid calculus formers increased progressively. However, when dental crowding was present, the relation between urea and this probability was not exhibited clearly; in fact, for the specific urea values between 20 and 50, subjects could belong to either group. This is to say, the presence of urea increases the probability of rapid calculus formation regardless of the presence of dental crowding. Dental crowding was not directly associated with the accumulation of calculus as the only etiological factor, but in combination with phosphorus and urea saliva levels.

## 5. Discussion

Analysis of the data obtained in the present study shows that periodontal diagnosis, dental crowding, levels of urea and phosphorus, and the presence of *S. mutans* bacteria are factors significantly related to the speed of calculus formation exhibited by an individual subject. The accumulation of dental calculus has been seen to be associated with the main periodontal diseases as bacterial plaque is deposited on tooth surfaces, which is the determining factor for developing these diseases [19]. The present results demonstrated that periodontitis at Stages III and IV or Grade C was associated with rapid formation of calculus, with statistical significance (*p* = 0.002). These findings support the observation that patients who accumulate large amounts of calculus generally present more severe periodontal problems.

Regarding biochemical components, statistically significant differences were only found in levels of phosphorus and urea, these being higher among rapid calculus-forming patients. Almerich et al., in a study on the composition of unstimulated saliva, placed normal phosphorus levels at 20.6 ± 8.5 mg/dL and urea levels at 45.8 ± 16.4 mg/dL [20]. These data agree with the present results obtained for rapid calculus formers, while slow calculus formers showed lower levels. 

In the present study, pH levels remained stable at different urea levels. In a study by Sissons et al., it was found that in an artificial mouth, as the concentration of urea increased, pH increased [21]. This could be due to the fact that the present study measured pH in saliva samples, while Sissons studied changes in pH in bacterial plaque.

The present results differed from findings by Mandel, who compared the saliva of heavy calculus-forming subjects with subjects who formed little calculus, finding that calcium concentrations in submaxillary saliva were significantly higher among calculus formers [22]. These differences may be due to the fact that in our study all saliva was collected, not just submaxillary saliva. Having reviewed the literature available on calculus accumulation, one line of investigation gives greater importance to the chemical components of bacterial plaque than those of saliva, as already observed in the case of pH. These articles focus on the biochemical components of bacterial plaque, leaving salivary components in second place. The work by Sissons observed increased urea concentrations in bacterial plaque, with increased pH due to the ammonia produced by ureolysis. This promotes calculus formation as the degree of calcium phosphate saturation in plaque increases [23].

Regarding microbiological parameters, of all the bacteria analyzed, only *S. mutans* showed a higher proportion in slow calculus-forming patients. According to the classic research conducted by Sidaway, it may be concluded that certain bacteria, whether alive or dead, have a calcifying capacity, especially *B. matruchotii,* which is the principal bacterial calcifier although not the only bacterium with this capacity [6]. The calcifying capacity of certain microorganisms has led researchers to investigate a range of bacteria. Moorer et al. performed a study to determine whether cariogenic streptococci exhibit this calcifying capacity. Calcium absorption for calcifying *S. mutans* C180-2, which has already been shown to be an acidogenic and cariogenic strain, was compared with *B. matruchotii*, a known calcifier. The mass of *S. mutans* presented a calcium concentration of 63 ± 11 mmol/kg, compared with 145 ± 61 mmol/kg in *B. matruchotii*. Calcifications were observed in some *S. mutans* cells [12]. Therefore, although *S. mutans* does have a capacity for calcification, it is much less than other microorganisms. 

Another line of investigation has set out to determine whether patients with a tendency for caries present less calculus accumulation due to bacterial competence. The present study observed that patients with large amounts of calculus presented lower concentrations of *S. mutans*. In 2010, Dahlén et al. published an article on microbiology in the presence of caries and calculus. The study took microbiological samples of bacterial plaque and saliva, analyzed them by means of DNA hybridization, and compared the bacteria present in two groups of patients, caries formers and calculus formers. It was found that there were significantly more bacteria among calculus formers than among caries formers; higher proportions were found of *Fusobacterium nucleatum, Prevotella intermedia, Prevotella negriscens* and *Prevotella tannerae*. Other bacteria were seen to be present sporadically: *Porphyromona gingivalis, Aggregatibacter actinomycetemcomitans, Tanerella forsythia* and *Campyloacter rectus*. Low frequencies were also observed for *S. mutans* and *Lactobacillus*. The authors concluded that there is not an inverse correlation between the proportion of caries and calculus, as many patients present as much caries as calculus and share many bacteria. They also pointed out that in their patient sample, no relation was found between the magnitude of calculus formation and the composition of bacterial plaque, but that calculus was associated with higher levels of salivary bacteria and with poor oral hygiene [24]. At the same time, studies that have administered anticalculus mouthwashes found that the microbiological parameters analyzed varied. For example, Sidaway observed that *S. mutans* had a much lower incidence at the start (10.8%) than after use of a placebo mouthwash (16.2%) or after the test mouthwash (21.6%). This change could be due to bacterial competence. When the proportions of *C. matruchottii* and *E. saburreum* decrease, the proportion of *S. mutans* increases [6]. These findings may be related to the present results, as patients treated with anticalculus mouthwash acquired a bacterial spectrum more similar to that described in the present study for slow calculus-forming patients. However, these variations do not always occur as a result of applying anticalculus mouthwashes. Fons-Badal et al., after using a pyrophosphate mouthwash, found no significant changes in the composition of bacterial plaque [25].

## 6. Conclusions

The speed of calculus formation appears to be significantly associated with patients’ individual characteristics and with biochemical and microbiological factors: Rapid calculus formation appears to be linked to patients with more severe periodontal diagnoses. Dental crowding is a much more frequent characteristic among rapid calculus formers.Regarding biochemical parameters, urea and phosphorus levels are higher among rapid calculus formers compared with slow calculus-forming control subjects.*Streptococcus mutans* is the only bacterium that appears to be related to the speed of calculus formation, as its prevalence is much greater among slow calculus formers.

## Figures and Tables

**Figure 1 jcm-09-00858-f001:**
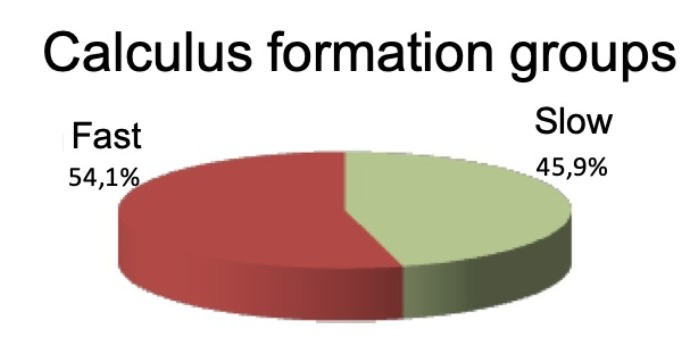
Pie chart showing sample distribution into two groups.

**Figure 2 jcm-09-00858-f002:**
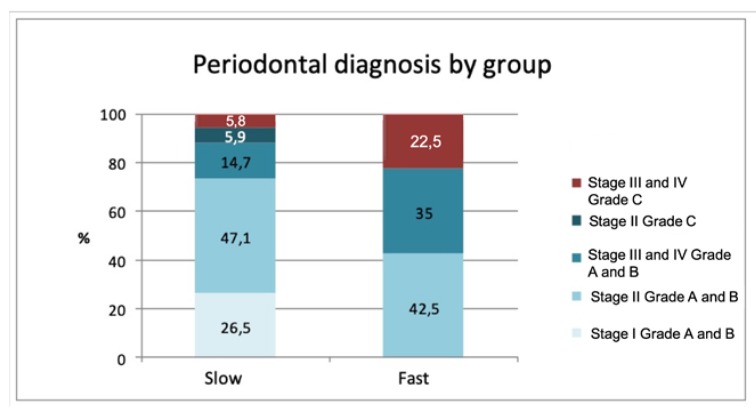
Graph showing the distribution of periodontal diagnoses in rapid and slow calculus formation groups.

**Figure 3 jcm-09-00858-f003:**
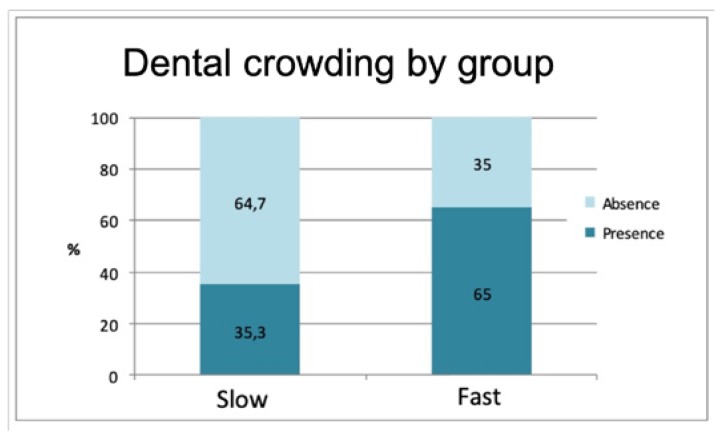
Distribution of dental crowding in rapid and slow calculus formers.

**Figure 4 jcm-09-00858-f004:**
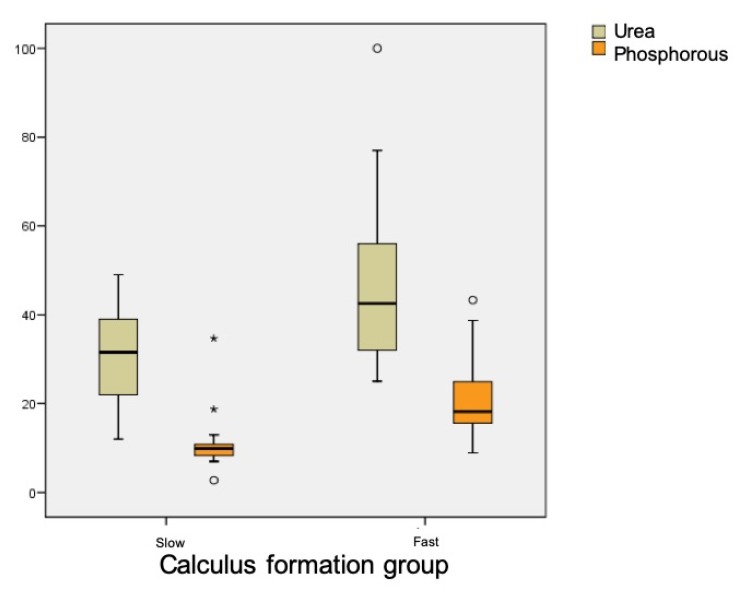
Distribution of urea and phosphorus in the two groups.

**Figure 5 jcm-09-00858-f005:**
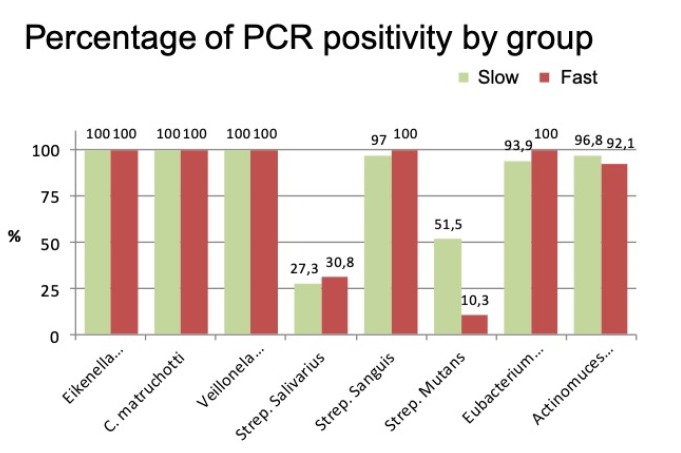
Percentage of bacteria analyzed in the groups of rapid and slow calculus formers.

**Figure 6 jcm-09-00858-f006:**
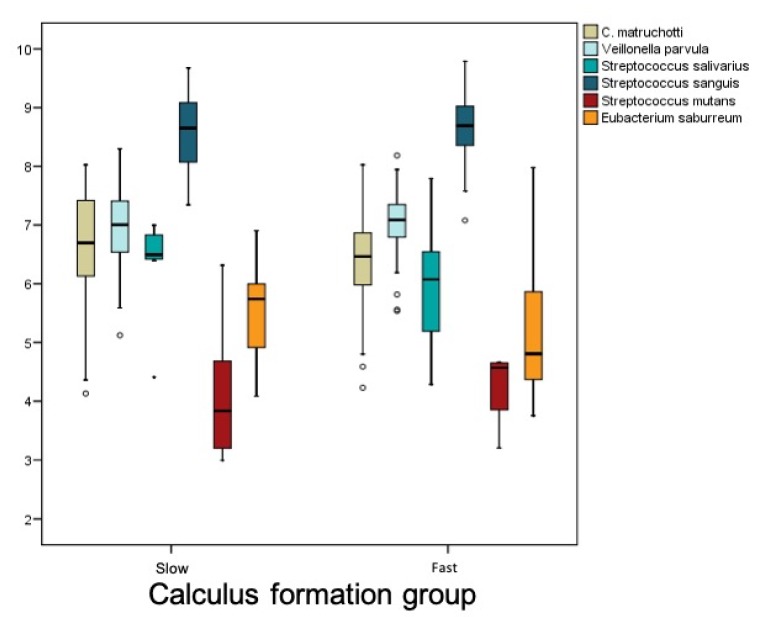
Bacteria distribution by group.

**Figure 7 jcm-09-00858-f007:**
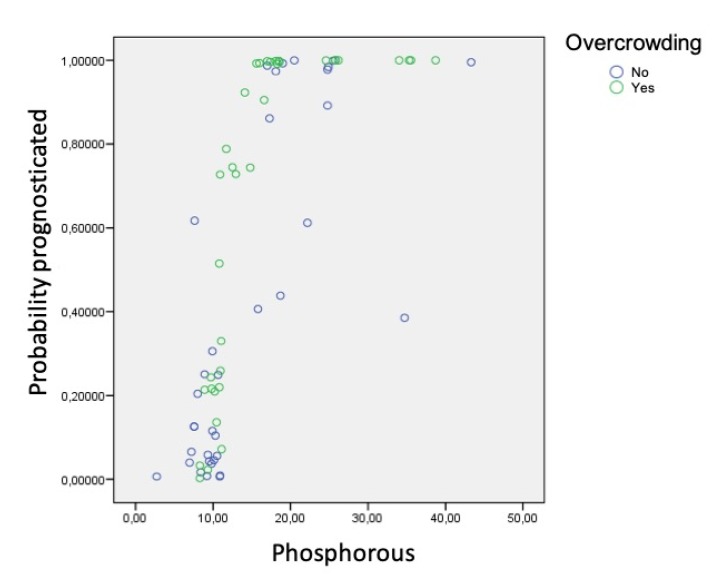
Relation between phosphorus levels and dental crowding.

**Figure 8 jcm-09-00858-f008:**
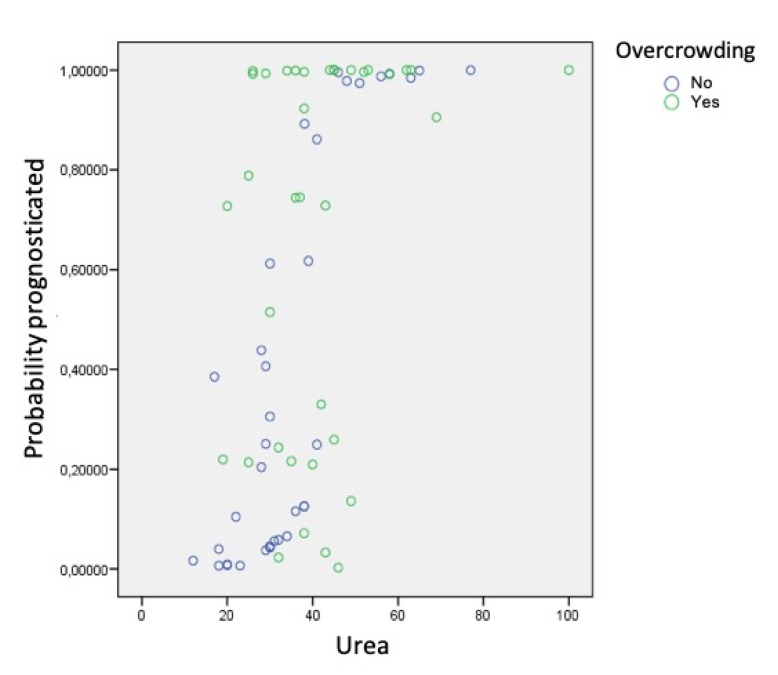
Relationship between levels of urea and dental crowding.

**Table 1 jcm-09-00858-t001:** Biochemical analysis results in the two calculus formation groups (slow and fast) and results of Mann–Whitney tests.

		Calculus Formation Group		*p*-Value (MW Test)
		Slow	Fast	
Unstimulated flow	Mean	0.57	0.52	0.444
pH	Mean	6.83	6.83	0.837
Urea	Mean	31.29	45.35	<0.001
Uric acid	Mean	2.74	3.47	0.083
Calcium	Mean	1.71	1.79	0.839
Phosphorus	Mean	10.32	20.45	<0.001
Sodium	Mean	8.97	9.69	0.477
Potassium	Mean	30.45	29.58	0.470
Chlorine	Mean	19.59	18.53	0.365

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
