# Peer review of "Analysis of Predisposing Factors for Rapid Dental Calculus Formation"

_jcm, 2020, doi:10.3390/jcm9030858_

Round 1

Reviewer 1 Report

To the Authors:

Thank you for your potential contribution to the Journal of Clinical Medicine. Please accept the comments and recommended revisions to improve the quality of your manuscript.

Sincerely,
Reviewer

Abstract

(Pg 1, line 11-12) Please reword this sentence:  by analyzing individual characteristics, and salivary, and microbiological parameters.

Introduction

(Pg 1, line 28-29) Is “biofilm” more appropriate than “plaque”?

(Pg 1, line 42-45) The purpose statement is very wordy. It may help to have an overarching purpose statement followed by specific aims.

Methods

(Pg 2, line 49) Can you please define the clinical parameters for maintenance phase?

(Pg 2, line 50) Do you mean “assessed” instead of divided?

(Pg 2, line 54) Please provide a general definition of the Volpe-Manhold Index.

(Pg 2, line 56) What plaque index are you referencing? (i.e. plaque index <1)

(Pg 2, line 58) Please provide the n-value next to the percentages.

(Pg 2, line 62) How was smoking habit assessed?

(Pg 2, line 65) How was overcrowding assessed and measured?

Results

(Pg 3, line 109) Please state how overcrowding was assessed.

(Pg 6, 162-164) Please change to past tense. Please correct spelling error in 2nd sentence of this paragraph (i.e. It).

Discussion

In general, please avoid using too much numerical data in this section. The actual numbers belong in the results section.

For the structure of the sentences, please reverse the order of the sentences in the paragraphs. You should state the findings of the present study and then follow with how the existing literature corroborates or contradicts your findings.

Conclusion

(Pg 8, line 237-245) Paragraphs should be composed on more than 1 sentence. These sentences should be combined into 1 paragraph or changed into bullet points.

References

Are there any more contemporary references to use? Most references are from 1970’s-1990’s.

Figures and Tables

These seem appropriate to display the data.

Author Response

Thank you very much for your recommendations for improving our manuscript. Please found below our responses.

REVISOR 1

Abstract

(Pg 1, line 11-12) Please reword this sentence:  by analyzing individual characteristics, andsalivary, and microbiological parameters.

Reply: Thank you. We have reword the sentence:

“The aim of this study was to analyze the individual characteristics, salivary and microbiological parameters between patients considered to be rapid calculus-formers compared with patients who form calculus slowly”.

Introduction

(Pg 1, line 28-29) Is “biofilm” more appropriate than “plaque”?

Reply: Thank you for pointing this out. We have changed it, “biofilm” instead of “placa”.

(Pg 1, line 42-45) The purpose statement is very wordy. It may help to have an overarching purpose statement followed by specific aims.

Reply: Thank you for your suggestion. We have reword the sentence dividing it into specific purposes:

“The aim of this study is to analyze differences between patients considered rapid calculus-formers compared with patients who form calculus slowly. The specific aims are: analyze individual characteristics between patients (age, sex, smoking, periodontal diagnosis, and overcrowding) and salivary and microbiological parameters in order to determine whether there are significant relations between these factors and calculus formation”.

Methods

(Pg 2, line 49) Can you please define the clinical parameters for maintenance phase?

Reply: Thank you for pointing this out. The clinical parameters that we take as a reference were: periodontal pockets £ 3 mm, plaque index (Silness and Löe) £ 1 and gingival index (Löe and Silness) £ 1.

(Pg 2, line 50) Do you mean “assessed” instead of divided?

Reply: Thank you for pointing this out. We have changed “assessed” to “evaluate”.

(Pg 2, line 54) Please provide a general definition of the Volpe-Manhold Index.

Reply: thank you for your comment, we have added the definition to the text:

 “Volpe-Manhold index is a method to quantify calculus in lingual surfaces of anterior lower teeth”.

(Pg 2, line 56) What plaque index are you referencing? (i.e. plaque index <1)

Reply: Thank you for pointing this out, it is Silness and Löe plaque index. We have added to the text:

 “All had antecedents of periodontitis, which had been treated so that patients were now in the maintenance phase (periodontal pockets £ 3 mm, plaque index (Silness and Löe) £ 1 and gingival index (Löe and Silness) £ 1)”.

(Pg 2, line 58) Please provide the n-value next to the percentages.

Reply: Thank you for your suggestion. We have added n-value:

“The study sample included 74 patients, of whom 40 (54.1%) were rapid formers and 34 (45.9%) slow formers, with a mean age of 51.3 ± 9.2 years, 53 (71.6%) women and 21 (28.4%) men. Most of the sample were not smokers (63.5%, n=47).”

(Pg 2, line 62) How was smoking habit assessed?

Reply: Thank you for pointing this out. The smoking habit was assessed in number of cigarettes per day:

“Firstly, patients’ individual characteristics were recorded: age, sex and smoking habit (number of cigarettes per day).”

(Pg 2, line 65) How was overcrowding assessed and measured?

Reply: Thank you for pointing this out, it was measured dichotomically: presence or absence. We have indicated it in the text:

“The presence of factors affecting bacterial plaque retention such as dental overcrowding (presence or absence) were also recorded.” 

Results

(Pg 3, line 109) Please state how overcrowding was assessed.

Reply: Thank you for pointing this out, it was measured dichotomically: presence or absence. We have indicated it in the text:

“The frequency of dental overcrowding (presence or absence) also exhibited significant difference between the groups (p = 0.011).”

(Pg 6, 162-164) Please change to past tense. Please correct spelling error in 2nd sentence of this paragraph (i.e. It).

Reply: thank you for your comment, we have changed it in the text:

“Dental overcrowding was not directly associated with the accumulation of calculus as the only etiological factor, but with its combination of the phosphorus and urea saliva levels.”

Discussion

In general, please avoid using too much numerical data in this section. The actual numbers belong in the results section.

For the structure of the sentences, please reverse the order of the sentences in the paragraphs. You should state the findings of the present study and then follow with how the existing literature corroborates or contradicts your findings.

 Reply: Thank you for your recommendation. We have reduced the numerical data and changed the order of the sentences to adapt it to the suggestion and understand it better.

Conclusion

(Pg 8, line 237-245) Paragraphs should be composed on more than 1 sentence. These sentences should be combined into 1 paragraph or changed into bullet points.

Reply: Thank you for your suggestion. We have changed it:

“The speed of calculus formation would appear to be significantly associated with patients’ individual characteristics and to biochemical and microbiological factors:

  1. Rapid calculus formation appears to be linked to patients with more severe periodontal diagnoses. Dental overcrowding is a much more frequent characteristic among rapid calculus formers.
  2. Regarding biochemical parameters, urea and phosphorous levels are higher among rapid calculus formers compared with slow calculus forming control subjects.
  3. Streptococcus mutans is the only bacteria that would appear to be related to the speed of calculus formation, as its prevalence is much greater among slow calculus formers.”

References

Are there any more contemporary references to use? Most references are from 1970’s-1990’s.

Reply: Thank you for pointing this out. It is an issue that has not been given importance in recent years in the literature, but we consider that is necessary continue studying it. That is why most of references are from 1970’s-1990’s.

Figures and Tables

These seem appropriate to display the data.

Reviewer 2 Report

In this study, the authors analyzed the differences in individual characteristics between rapid calculus-formers and slow calculus-formers, and found that the speed of calculus formation was significantly associated with patients’ individual characteristics and to biochemical and microbiological factors. The results are potentially interesting, but I have some minor concerns on the experimental design and description in the manuscript.

  1. How were the patients diagnosed as advanced aggressive, moderate aggressive and so on? The definition of these classification should be described in Materials and methods.
  2. How overcrowding was defined?
  3. Page 4, Line 117-119: The difference in uric acid was mentioned in this paragraph, but no results related to uric acid was shown in Figure 4.
  4. Page 5, Line 125-127: The prevalence of bacteria was mentioned in this paragraph, but the title of Figure 5 is “Percentages of bacteria”. Which is it, the prevalence of bacteria or percentages of bacteria?
  5. Page 5, Line 131: What does “slow formers: 58282.6CFU; fast formers: 4801.8CFU” mean? Mean value? Are these results shown in Figure 6?
  6. Page 7, Line 189-191: Mandel et al. found that the difference in calcium concentration of submaxillary saliva, but not parotid saliva.  The whole saliva seemed to be collected in this study.
  7. Page 7, Line 217: “Porfhyromonas”, “Porphyromonas”.
  8. Page 8, Line 219: “Lactobacilus”, “Lactobacillus”
  9. The Discussion was divided into too many paragraphs.

Author Response

Thank you very much for your recommendations for improving our manuscript. Please found below our responses.

Comments and Suggestions for Authors

In this study, the authors analyzed the differences in individual characteristics between rapid calculus-formers and slow calculus-formers, and found that the speed of calculus formation was significantly associated with patients’ individual characteristics and to biochemical and microbiological factors. The results are potentially interesting, but I have some minor concerns on the experimental design and description in the manuscript.

How were the patients diagnosed as advanced aggressive, moderate aggressive and so on? The definition of these classification should be described in Materials and methods.

Reply: Thank you for your suggestion. We refer to the classification of periodontal diseases of Armitage (1999). We have added it in the text:

“A periodontal examination was performed to assess periodontitis stability and make an initial diagnosis of periodontal disease according to the classification of Armitage in 1999.”

How overcrowding was defined?

Reply: Thank you for pointing this out, it was measured dichotomically: presence or absence. We have indicated it in the text:

“The frequency of dental overcrowding (presence or absence) also exhibited significant difference between the groups (p = 0.011).”

Page 4, Line 117-119: The difference in uric acid was mentioned in this paragraph, but no results related to uric acid was shown in Figure 4.

Reply: Thank you for pointing this out. It has not been shown in Figure 4 because the results of uric acid were not statistically significant. We decided only to highlight in Figure 4 the statistically significant results.

Page 5, Line 125-127: The prevalence of bacteria was mentioned in this paragraph, but the title of Figure 5 is “Percentages of bacteria”. Which is it, the prevalence of bacteria or percentages of bacteria?

Reply: Thank you for pointing it out the error. Is “Percentages of bacteria”, not “Prevalence of bacteria”.  We have changed it in the text:

“Figure 5. Percentage of bacteria analyzed in groups of rapid and slow calculus formers.”

Page 5, Line 131: What does “slow formers: 58282.6CFU; fast formers: 4801.8CFU” mean? Mean value? Are these results shown in Figure 6?

Reply: Thank you for pointing this out. CFU refers to colony-forming-units, sorry for the lack of clarification in the text, has been added:

“A notable difference between the groups was only found for S. mutans, which was higher in the group of slow calculus formers (slow formers: 58282.6CFU (colony-forming-units)”

Page 7, Line 189-191: Mandel et al. found that the difference in calcium concentration of submaxillary saliva, but not parotid saliva.  The whole saliva seemed to be collected in this study.

Reply: Thank you for your suggestion. in our study all saliva is collected, not just submaxillary saliva. We have added this interesting clarification:

 “These differences may be due to de fact that in our study all saliva is collected, not just submaxillary saliva”

Page 7, Line 217: “Porfhyromonas”, “Porphyromonas”.

Reply: Thank you for pointing this out. We have changed it.

Page 8, Line 219: “Lactobacilus”, “Lactobacillus”

Reply: Thank you for pointing this out. We have changed it.

The Discussion was divided into too many paragraphs.

Reply: Thank you for your suggestion. We have unified paragraphs by topic to improve this point.

Round 2

Reviewer 1 Report

To the authors:

Thank you for making the recommended revisions. However, you will need to add references to the reference list for all the indices used in your manuscript. It is not acceptable to refer to the source in the text and not add the reference to the list of references. This is a major revision.

Each of the indices should be explained. This may not be common knowledge for all readers. What does plaque index of <1 mean? What does ginival index <1 mean? What does Volpe-Manhold index mean?

Please define the reference for overcrowding? How can it be defined as absence or presence? Does it involve 1 tooth or more than 1 tooth? Who defined it like this?

For periodontal classification, why are you using the 1999 APA classifications? Why didn't you use the 2017 APA classification? I would recommend using both and add the reference to the list of references. It is not acceptable to refer to the source in the text and not add the reference to the list of references.

Sincerely,
Reviewer

Author Response

Thank you very much for your new recommendations. Please found below our responses.

Thank you for making the recommended revisions. However, you will need to add references to the reference list for all the indices used in your manuscript. It is not acceptable to refer to the source in the text and not add the reference to the list of references. This is a major revision.

Each of the indices should be explained. This may not be common knowledge for all readers. What does plaque index of <1 mean?

Reply: Thank you for your suggestion. We have added the definition and reference to the text:

“Silness and Löe plaque index £ 1 (plaque is not visible but can be wiped off with a explorer10)”.

Silness J, Löe H. Periodontal disease in pregnancy. II. Correlation between oral hygiene and periodontal condition .Acta Odontol Scand. 1964 Feb;22:121-35.

What does ginival index <1 mean?

Reply: Thank you for your suggestion. We have added the definition and reference to the text:

“Löe and Silness gingival index £ 1 (mild inflammation, slight color change, mild alteration of gingival surface, no bleeding11

Löe H. The Gingival Index, the Plaque Index and the Retention Index Systems. J Periodontol. 1967 Nov-Dec;38(6):Suppl:610-6.

What does Volpe-Manhold index mean?

Reply: Thank you for your suggestion. We have added the definition and reference to the text:

“Volpe-Manhold index is a method to quantify calculus making measurements in lingual surfaces of anterior lower teeth using the periodontal probe13

Volpe AR, Manhold JH y Hazen SP. (1965) “In Vivo Calculus Assessment: Part I. A method and its examiner reproducibility.” en J Periodontol. 36:292-8.

Please define the reference for overcrowding? How can it be defined as absence or presence? Does it involve 1 tooth or more than 1 tooth? Who defined it like this?

Reply: Thank you for pointing this out. We have changed the word “overcrowding” by “dental crowding”. After consulting a lot of bibliography we have considered that the term “dental crowding” is more correct and the text is going to be understand  better.

“Dental crowding can be defined as a discrepancy between tooth and jaw sizes that results in malposition and/or rotation of teeth15. Its presence or absence was registered only in the 5th sextant because it is where we were going to measure the index. “

Howe RP, McNamara JA Jr, O’Connor KA. An examination of dental crowding and its relationship to tooth size and arch dimension. Am J Orthod. 1983;83:363–373. doi: 10.1016/0002-9416(83)90320-2.

For periodontal classification, why are you using the 1999 APA classifications? Why didn't you use the 2017 APA classification? I would recommend using both and add the reference to the list of references. It is not acceptable to refer to the source in the text and not add the reference to the list of references.

We had used the Armitage classification of 1999 because at the time of sample collection it was the one in force. But given after your suggestion, we have re-classified the patients and redone the tables and results, adapting them to the new classification (2017 APA classification) published in 2018.

“A periodontal examination was performed to assess periodontitis stability and make an initial diagnosis of periodontal disease according to the classification of Caton et al 201814”.

“This relation between rapid formers and the type of periodontitis they presented is shown in Figure 2, whereby Stages III and IV and grade C was associated with rapid calculus formation (advanced and aggressive periodontitis according to the classification of Armitage 199918).”

Caton JG, Armitage G, Berglundh T, Chapple ILC, Jepsen S, Kornman KS, Mealey BL, Papapanou PN, SanzM, TonettiMS.A new classification scheme for periodontal and periimplant diseases and conditions - Introduction and key changes from the 1999 classification. J Clin Periodontol. 2018 Jun;45 Suppl 20:S1-S8.

Armitage GC. Development of a classification system for periodontal diseases and conditions. Ann Periodontol. 1999 Dec;4(1):1-6.
